# Primary Outcomes of a Randomized Controlled Crossover Trial to Explore the Effects of a High Chlorophyll Dietary Intervention to Reduce Colon Cancer Risk in Adults: The Meat and Three Greens (M3G) Feasibility Trial

**DOI:** 10.3390/nu11102349

**Published:** 2019-10-02

**Authors:** Andrew D. Frugé, Kristen S. Smith, Aaron J. Riviere, Wendy Demark-Wahnefried, Anna E. Arthur, William M. Murrah, Casey D. Morrow, Robert D. Arnold, Kimberly Braxton-Lloyd

**Affiliations:** 1Department of Nutrition, Dietetics and Hospitality Management, Auburn University, Auburn, AL 36849, USA; kss0034@auburn.edu (K.S.S.); ajr0042@auburn.edu (A.J.R.); 2Department of Nutrition Sciences, University of Alabama at Birmingham, Birmingham, AL 35294, USA; demark@uab.edu; 3Department of Food Science and Human Nutrition, Division of Nutritional Sciences, University of Illinois at Urbana-Champaign, Champaign, IL 61801, USA; aarthur@illinois.edu; 4Department of Educational Foundations, Leadership, and Technology, Auburn University, Auburn, AL 36849, USA; wmm0017@auburn.edu; 5Department of Cell, Developmental and Integrative Biology, University of Alabama at Birmingham, Birmingham, AL 35294, USA; caseym@uab.edu; 6Department of Drug Discovery and Development, Auburn University Harrison School of Pharmacy, Auburn, AL 36849, USA; rda0007@auburn.edu; 7Department of Pharmacy Services, Auburn University Harrison School of Pharmacy, Auburn, AL 36849, USA; lloydkb@auburn.edu

**Keywords:** chemoprevention, colon cancer, diet, green leafy vegetables, red meat

## Abstract

Preclinical and observational research suggests green leafy vegetables (GLVs) may reduce the risk of red meat (RM)-induced colonic DNA damage and colon cancer (CC). We sought to determine the feasibility of a high GLV dietary intervention in adults with an increased risk of CC (NCT03582306) via a 12-week randomized controlled crossover trial. Participants were randomized to immediate or delayed (post-4-week washout) intervention groups. During the 4-week intervention period, participants were given frozen GLVs and counseled to consume one cooked cup equivalent daily. The primary outcomes were: accrual—recruiting 50 adults in 9 months; retention—retaining 80% of participants at completion; and adherence—meeting GLV intake goals on 90% of days. Adherence data were collected twice weekly and 24-h dietary recalls at each time point provided nutrient and food group measures. The Food Acceptability Questionnaire (FAQ) was completed to determine acceptability. On each of the four study visits, anthropometrics, stool, saliva, and blood were obtained. Fifty adults were recruited in 44 days. Participants were 48 ± 13 years of age, 62% female, and 80% Caucasian, with an average BMI at screening of 35.9 ± 5.1. Forty-eight (96%) participants were retained and completed the study. During the intervention phase, participants consumed GLVs on 88.8% of days; the adherence goal of one cup was met on 73.2% of days. Dietary recall-derived Vitamin K and GLVs significantly increased for all participants during the intervention periods. Overall satisfaction did not differ between intervention and control periods (*p* = 0.214). This feasibility trial achieved accrual, retention and acceptability goals, but fell slightly short of the benchmark for adherence. The analysis of biological specimens will determine the effects of GLVs on gut microbiota, oxidative DNA damage, and inflammatory cytokines.

## 1. Introduction

Over 51,000 men and women in the United States will die from colon cancer (CC) in 2019 [1]. The same number of new CC diagnoses could be prevented each year through meeting the American Cancer Society’s Diet and Physical Activity Cancer Prevention Guidelines [2]. Obesity alone increases the risk of CC by 33% and is associated with increased CC mortality [3]. Red and processed meat consumption is associated with increased CC risk [4], most often in the context of dietary patterns, which juxtapose “Western” and “prudent” diets [5,6]. It is clear that the Western diet increases the risk of CC, but it is important to determine whether the risk could be reduced by adding prudent foods to a Western diet. A recent meta-analysis of 24 case control studies and 11 prospective cohort studies (*n* = 1,295,063 men and women) found an 18% reduction in the risk of CC in the groups consuming the highest levels of cruciferous and green leafy vegetables (GLVs) [7]. Risk reduction with diets high in green leafy vegetable consumption remained significant regardless of whether pooled studies controlled for meat and/or total energy intake [7]. 

Preclinical studies have established that dietary heme is the primary carcinogen in red meat, with minimal effects from N-nitroso compounds [1], heterocyclic amines [2,3], and polycyclic aromatic hydrocarbons [4,5]. Heme is modified in the intestinal lumen and induces necrosis in epithelial cells [6], resulting in compensatory hyperproliferation in colonic crypts (Figure 1) [7]. Interestingly, spinach and chlorophyll prevent cytotoxicity and damage to colonocytes in vivo by the binding of heme between chlorophyll molecules [8]. A longitudinal analysis of the Netherlands Cohort Study (*n* = 120,852) supports this outcome, with men consuming the highest molar ratios of dietary heme to chlorophyll experiencing increased colon cancer risk (Relative Risk (RR) 1.43, 95% confidence interval (CI) 1.03–1.97) [9]. Heme-induced genotoxicity of the colon is prevented by chlorophyll in rodent models [6,10,11,12], an important finding that has not been tested in clinical trials though is supported by epidemiological observation. 

We recently found that among 990 adults living throughout the United States, the unwillingness to remove red meat from the diet was highly prevalent among respondents, though dislike for green leafy vegetables was rare [13]. These results suggest that encouraging the general public to increase green leafy vegetables consumption may be a more effective CC risk reduction strategy than discouraging red meat intake alone. In a first step toward exploring the effects of green leafy vegetable intake on systemic DNA damage and inflammatory markers, we first undertook a 12-week randomized controlled crossover study to determine the feasibility of translating this intervention in adults with an increased risk of colon cancer. 

## 2. Materials and Methods

### 2.1. Study Design and Aims

This trial utilized a crossover design to assess the feasibility of a dietary intervention in adults with increased risk of CC. This study was approved by the Auburn University Institutional Review Board, protocol # 18-180 EP 1806. The study was pre-registered on ClinicalTrials.gov (NCT03582306) and was conducted in accordance with the Declaration of Helsinki. 

#### 2.1.1. Primary Aim

We sought to determine the feasibility of a randomized controlled crossover trial employing high green leafy vegetable consumption in adults with an increased risk of colon cancer. Feasibility was measured by accrual, retention, and adherence. The accrual target was enrollment of 50 subjects within 9 months. The retention target was to retain 90% of participants at crossover and 80% at completion. The adherence target was for participants to meet a GLV intake goal (1 cooked cup daily) on 90% of days.

#### 2.1.2. Secondary Aims

The acceptability of the intervention was compared to the control period using the Food Acceptability Questionnaire (FAQ) [14,15] and dietary recall data. In addition, and in an effort to obtain pilot data that are relevant to carcinogenesis, changes in circulating DNA damage, inflammatory cytokines and the gut microbe composition will be explored; however, in this paper, which focuses on feasibility outcomes, these data are not reported. 

### 2.2. Participant Recruitment and Informed Consent

#### 2.2.1. Recruitment

Participants were recruited through the Auburn University Pharmaceutical Care Center via an open email solicitation with periodic follow-up from July to September 2018. Participants were directed to the study website which provided the study overview, eligibility criteria, consent form, link to clinicaltrials.gov, and a Qualtrics eligibility survey link. Fifteen Diet History Questionnaire II questions (DHQ-II) [16] were used to quantify habitual intake of green leafy vegetables and red meat over the past thirty days. Self-reported demographics, height, and weight were also provided. Every person who completed the survey was contacted and thanked for their interest. Those who were ineligible were invited to participate in a separate study. Those who were eligible and interested were scheduled for a consent appointment after study staff provided a description of the study and answered any potential questions via telephone. 

#### 2.2.2. Eligibility

Individuals were eligible for participation if the following criteria were met: (1) Currently consume less than two servings of green leafy vegetables per day; (2) Currently consume five or more servings of red meat per week; (3) Have a BMI > 30 kg/m^2^; (4) Agree to not change dietary supplements over the course of the study; (5) Are willing to comply with the dietary regimen over the course of the study; (6) Are able to store and cook frozen green leafy vegetables (at minimum, a freezer and microwave); and (7) are able to speak and read English. Exclusion Criteria included previous diagnosis of CC and use of the following microbiome-altering drugs within the last four weeks: systemic antibiotics, corticosteroids, immunosuppressive agents, or commercial probiotics [17]. 

### 2.3. Baseline

After eligible participants were initially contacted, written consent was obtained, and participants were given a stool collection kit with instructions for obtaining the sample. Participants reported dietary information through two 24-h dietary recalls on the days immediately preceding the sample collection, collected over the phone or via email. Participants fasted for 8–12 h prior to baseline assessment, at which time they submitted their fecal sample, underwent phlebotomy, and completed questionnaires. Patients were provided with an additional stool collection kit for the 4-week follow-up appointment.

### 2.4. Randomization

After completing all baseline procedures, participants were randomized in blocks of 4, which were stratified by gender due to potential differences in fecal microbiome [18]. All participants received the intervention; the order in which it was received was randomly generated via an online computer program [19]. 

### 2.5. Interventions

Participants received the intervention period during the first four (immediate intervention group) or last four weeks (delayed intervention group) of the study. During the intervention period, a goal of 1 cup cooked dark leafy green vegetables was prescribed, and participants were instructed to consume at least ½ cup cooked GLVs during the same meal they consumed red meat. During the intervention period, participants received a variety of frozen GLVs, including spinach, kale, collards, mustard greens, and turnip greens. This produce was processed by large-scale producers and purchased for study use from retail outlets. Quantities provided on a weekly or bi-weekly basis were sufficient to provide the goal amounts assuming cooking reduction. Frozen vegetables were provided because flash-freezing minimizes nutrient loss and prevents chlorophyll degradation [20]. Though the chlorophyll content varies between these species, it is exponentially higher than other green vegetables in the brassica family. Participants were instructed to consume cooked vegetables, rather than raw, to increase the bioavailability of chlorophyll [21] and reduce the volume of the needed amount of vegetable. Participants were provided with a recipe book detailing preparation instructions, as well as various recipes, for each GLV in order to reduce barriers to consumption. 

After each 4-week period, participants completed questionnaires, provided a stool sample, and underwent phlebotomy. Study staff obtained two 24-h dietary recalls on two of the four days preceding stool sample collection at each time point. The behavioral framework of this intervention utilized Social Cognitive Theory [22] as the basis for behavior change during the intervention period. Participants increased self-efficacy of dietary adherence by setting goals and logging daily food intake (self-monitoring), which was assessed in twice weekly counseling sessions with a registered dietitian (RD).

### 2.6. Measures/Time Points Data Collection

Details regarding measurements at each time point are reported in Table 1. Two 24-h dietary recalls were obtained by an RD or dietetics student and entered into the Automated Self-Administered 24-Hour Dietary Assessment tool (ASA24), which uses the United States Department of Agriculture Food and Nutrient Database for Dietary Studies (FNDDS) to provide values for 195 nutrients, nutrient ratios and other food components [23]. Because changes in physical activity could affect microbiome composition [24], self-reported physical activity was assessed using the International Physical Activity Questionnaire (IPAQ) at each time point [25]. Height and weight were measured using standard procedures and used to calculate body mass index (BMI). Waist and hip circumferences were measured using a standard tape measure at each time point. Body composition was analyzed using a handheld Body Impedance Analysis (BIA) instrument. Fecal samples were collected by participants using commode specimen collectors and sterile collection tubes, which were immediately stored in a home freezer until the time of their appointments. Blood was drawn by a trained phlebotomist and serum and plasma were separated and frozen at −80 °C until analysis. 

Diet acceptability was assessed with the Food Acceptability Questionnaire (FAQ) [14,15] after each intervention period to investigate differences in acceptability between control and intervention diets. The FAQ utilizes 10 questions on a 7-point Likert scale to assess enjoyment, difficulty of preparation and adherence, and satisfaction with the diet protocol [26].

The previously validated Dietary Habits and Colon Cancer Beliefs Survey (DHCCBS) was distributed and collected at baseline and final appointment to assess changes in beliefs and attitudes related to perceived CC risk. DHCCBS contains 13 questions utilizing the Health Belief Model to assess perceived CC susceptibility and severity, and related dietary barriers, benefits, and calls to action [27]. 

### 2.7. Hypothesis/Power/Statistical Analysis

The primary outcomes of this feasibility study were accrual, retention, and adherence. Power analysis was originally based on adherence. Setting alpha = 0.05, beta = 0.20, and *n* = 44 (assuming 10% attrition), and 90% adherence will result in a Cohen’s D = 0.80, which is a large effect size [28].

Statistical Analysis: Statistical analyses were conducted in SPSS 24.0 (IBM Corp. Released 2016. IBM SPSS Statistics for Windows, Version 24.0. Armonk, NY, USA: IBM Corp.) Descriptive statistics were obtained for study participants and non-participants, which were compared using independent sample *t*-tests for continuous variables and chi-square tests for categorical variables. The change in dietary intake variables and FAQ scores between time points were assessed with paired sample *t*-tests within each study arm. 

## 3. Results

### 3.1. Feasibility Outcomes

A total of 178 adults completed the screening survey and were assessed for eligibility. Fifty eligible adults were recruited and enrolled in the study in 44 days. Participant characteristics are compared to non-participants in Table 2. Those enrolled in the study had a higher BMI and less racial diversity than those not enrolled. Furthermore, participants were not significantly older and consumed significantly more red meat (RM) servings per week (*p* = 0.05). Participant recruitment and retention is outlined in Figure 2. One participant withdrew consent due to aggravation of diverticulitis, and one was lost to follow-up after illness—both were in the immediate group. Forty-eight (96%) participants were retained and completed the study. 

A total of 463 adherence days were attained from participants during the intervention phase. During the intervention phase, participants consumed some GLVs on 88.8% of days and the adherence goal of 1 cooked cup/day was met on 73.2% of days. 

Total nutrients and food groups were ascertained from each participants’ 24-h dietary recalls. Values from each of the two recalls at each time point were averaged and reported for key variables in Table 3. For the immediate group, dietary Vitamin K and the Dark Green Vegetables food group increased significantly during the intervention period (*p* = 0.009; *p* = 0.011, respectively) and remained constant during the control period. Red meat consumption was relatively unchanged for the immediate group throughout the study. For the delayed group, key diet variables did not change during the control period. However, Vitamin K (*p* = 0.001, *p* = 0.006, *p* = 0.026, respectively), Dark Green Vegetables and Red Meat all increased significantly during the intervention period.

### 3.2. Acceptability Outcomes

Within and between group comparisons are reported for each FAQ question and the Total FAQ score in Figure 3. When comparing post-intervention and control responses for each group, a statistically significant difference was observed in the immediate group in response to several questions. With regard to liking of the intervention foods (questions 1–3), mean scores were lower for the intervention period compared to the control period. Overall satisfaction (question 10) was rated lower during for the intervention period, with trends (*p* < 0.1) in lower scores for effort to stay on the diet (question 8) and after-meal satisfaction (question 9). Subsequently, the sum FAQ score was lower for the intervention period in the immediate group (*p* = 0.006).

For the delayed group, fewer differences were observed between the intervention and control periods. Participants in the delayed group reported lower scores for ease of purchasing the foods (question 6), maintaining the diet at restaurants (question 7), and effort to stay on the diet (question 8), which suggests these participants were more likely to travel or not cook the study foods. When all participants were combined, the sum FAQ score was lower for the intervention period compared to control, but overall satisfaction (question 10) was not different between groups.

## 4. Discussion

In this randomized controlled crossover trial, we found that a high green leafy vegetable dietary intervention is feasible in adults at a higher risk of colon cancer. 

The accrual target was met rapidly and at a higher efficiency than expected since we observed an accrual rate of 28% in 44 days. Retention, above the goal of 80%, was also acceptable. Diverticular disease is a heterogeneous disease which is often exacerbated by dietary fiber, so we chose not to exclude individuals with this condition, especially since they are at increased risk of developing CC or colorectal cancer (CRC) [29]. Because this proved problematic for one participant that was unable to continue the intervention, we will exclude participants such as these from future studies, as well as those with any other known gastrointestinal diagnoses such as Irritable Bowel Disease, Crohn’s Disease, Celiac, Ulcerative Colitis, etc. The second participant that was lost to follow up may have been retained if we had chosen to keep the original timeline rather than postpone the first follow-up appointment due to antibiotics. Another potential solution would have been to utilize a post-randomization exclusion since current or recent antibiotics were an exclusion criterion.

The adherence goal was not achieved, although participants did report partial and sporadic adherence. Participants with low adherence often reported overnight travel as a major barrier to consuming GLVs, especially when they were outside of the southeastern US and unable or unwilling to seek out large quantities of cooked GLVs. Some participants also reported that they did not have time to properly prepare the greens. Many of the GLV packages suggest that they must be cooked for at least 30 min, so individuals who do not normally cook were especially unlikely to spend the time needed for food preparation. A small proportion (*n* = 3) chose to meet GLV goals by consuming them in smoothies. This sample is likely too small to determine whether mode of consumption differed physiologically. Data from the 24-h dietary recalls are more promising as Vitamin K intake levels increased 2- and 3-fold during the intervention periods. Similarly, the Dark Green Vegetables group increased from approximately one quarter of a cup per day to 0.85 cups, reflecting a statistically significant increase in GLV intake. A recent meta-analysis of 27 weight loss interventions found that average adherence in those studies in a similar population was 60.5% [30]. Only 53.4% of 23,841 cancer survivors participating in diet and exercise interventions met 75% adherence goals, indicating that even highly motivated populations often fall short of much more conservative adherence goals [31]. These findings indicate our 90% adherence goal was unrealistic and our observed 73.2% adherence rate was comparable to other recent trials. 

Results from our study are supported by recent interventions in cancer prevention and control. Diets high in fruits and vegetables may lower cancer risk, specifically CRC [32]. Furthermore, cruciferous vegetables and their active components have chemopreventive effects through various mechanisms [33]. Active components, such as folate, chlorophyll, or fiber, interact with receptors and transcription factors to provide beneficial protective effects throughout the gut [34]. Moreover, a recent meta-analysis of 21 cohort and case-control studies indicated a significantly lower risk of CC in men and women with the highest levels of dietary fiber consumption [35]. 

Most recently, Crowder et al. reported that head and neck cancer survivors were receptive to and adherent of a 12-week intervention including weekly correspondence and counseling from a Registered Dietitian. In this pilot study, the intervention group displayed an mean increase in GLV intake by 5.5 cups and cruciferous vegetables by 3.5 cups per week [36]. Pierce et al. investigated the feasibility of a high vegetable diet in breast cancer survivors to prevent recurrence. The intervention group received telephone counseling sessions and monthly cooking classes and were encouraged to consume five or more servings of vegetables per day. The intervention group consumed significantly more servings of vegetables per day than the control group at 6 and 12 months. Results from this study indicate breast cancer survivors are able to adopt a high-vegetable, low-fat, and increased fiber diet [37]. Both of these studies indicate dietary counseling is effective in improving diet quality as a secondary prevention strategy.

Few studies have included a dietary component for CC/CRC primary prevention. The North Carolina Strategies for Improving Diet, Exercise, and Screening (NC STRIDES) aimed to promote health through diet, physical activity, and CRC screenings among cancer-free individuals and CRC survivors. This study indicated efficacy of dietary interventions to increase fruit and vegetable consumption in adults. Both the general population and CRC survivors increased fruit and vegetable intake when participants received telephone motivational interviewing combined with printed materials [38]. Baker and colleagues reported increased fruit and vegetable intake in individuals attending colorectal cancer screenings with simple written messages about modification of diet for cancer-protective effects. These dietary behaviors were significantly higher in adults receiving the written message compared to control group. This study also indicates cancer screenings as an important point of contact for chemopreventive dietary interventions [39].

Because this is a feasibility trial, several limitations must be acknowledged. First, self-reported data were used for adherence measures, and as such, we could not assess whether reported behaviors were comparable on non-reported days. Although a variety of cooking methods were utilized, these differences were not captured. Data were only collected on preparation methods regarding cooked, raw, or in smoothies. These differences may account for variability in chlorophyll bioavailability. We did have several participants who experienced unexpected health benefits from regular GLV consumption such as relief from gastroesophageal reflux (*n* = 2) and decreased lactose sensitivity (*n* = 1). This led to persistence in high GLV intake of immediate group participants for the duration of the study, including washout and control periods. The addition of Plasma Vitamin K1 to biomarker analysis will provide an objective measure of GLV consumption and will provide further clarity with regard to dose response in DNA damage and inflammatory biomarkers. Another unintended consequence of this study was that most participants spontaneously reduced RM consumption at the outset of the study. The RM questions from our screener were validated, though may have been overly sensitive. Nonetheless, future studies will need to ensure concomitant RM consumption to verify whether GLVs are actually providing the proposed heme-binding benefit. 

Finally, our study population was unique. Participants were recruited through Auburn University’s Pharmaceutical Care Center, which serves all faculty and staff. However, our sample was overly representative of faculty and other highly educated men and women, thus these results may not be generalizable to the public. We were, however, able to recruit a significant proportion of African-American participants (*n* = 10, 20%) and males (*n* = 19, 38%), but greater efforts will be required to better represent these groups, who are still at an elevated risk and might benefit the most from preventive interventions [40]. 

Future trials should investigate varied quantities of GLVs to identify the dose needed to maximize benefits. Similarly, controlled studies should determine the quantities of a GLV and RM diet where carcinogenic exposure is minimized. Also, the timing of the greens and red meat consumption is important for determining whether greens provide most benefit when consumed with red meat by actually binding to the heme, or if the benefits can be obtained as long as they are consumed within a few hours of each other. 

## 5. Conclusions

Results from this randomized control trial indicate a high green leafy vegetable dietary intervention is feasible in adults at high risk of colon cancer. The analysis of biological specimens will determine physiological effects and shape future investigations. 

## Figures and Tables

**Figure 1 nutrients-11-02349-f001:**
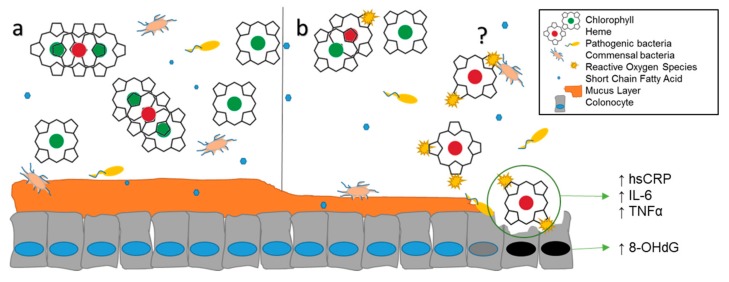
Anticipated effects of high vs. low green leafy vegetable consumption in the colon. (**a**) In the lumen, chlorophyll binds heme, preventing cytotoxicity. (**b**) Heme is easily oxidized in the absence of chlorophyll; it is unknown if microbes, their metabolites, or both react with heme. Degradation of the mucin layer increases the susceptibility of epithelial cells to pathogens and oxidative stress, resulting in elevated systemic inflammation. Necrosis leads to compensatory hyperproliferation and DNA damage.

**Figure 2 nutrients-11-02349-f002:**
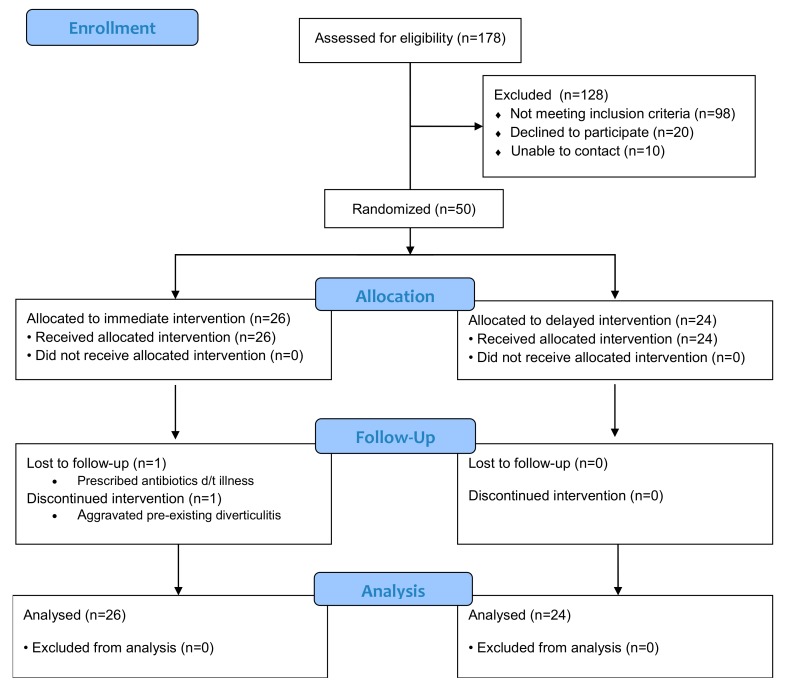
Consolidated Standards of Reporting Trials (CONSORT) Flow Diagram.

**Figure 3 nutrients-11-02349-f003:**
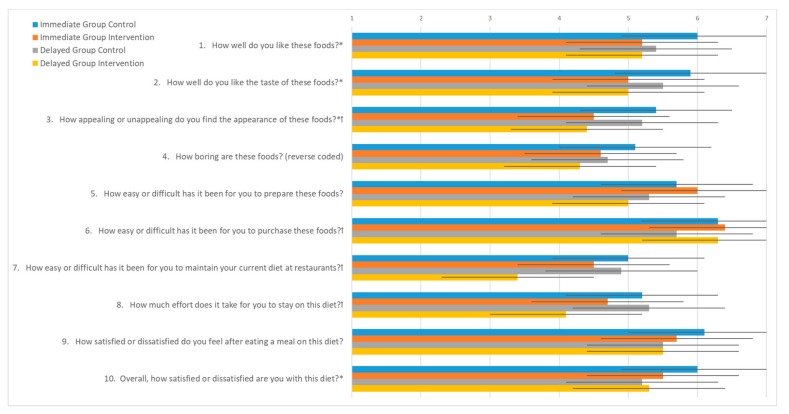
Food Acceptability Questionnaire with 7-point Likert scale responses from immediate and delayed intervention groups after control and intervention periods during a randomized controlled crossover high green leafy vegetable dietary intervention. Response options range from 1 (not at all/easy) to 7 (extremely (easy)). * indicates a significant difference (*p* < 0.05) between control and intervention FAQ responses for Immediate Group. ꝉ indicates a significant difference (*p* < 0.05) between control and intervention FAQ responses for Delayed Group.

**Table 1 nutrients-11-02349-t001:** Timeline and measures for a randomized controlled crossover high green leafy vegetable dietary intervention.

	Base	4 Weeks	8 Weeks	12 Weeks
Primary Outcome:				
Feasibility—accrual, adherence, retention	X	X	X	X
Secondary Outcomes:				
24-h dietary recalls—diet composition	X	X	X	X
International Physical Activity Questionnaire	X	X	X	X
Anthropometrics—height (only at baseline), weight, body mass index (BMI), waist and hip measurements	X	X	X	X
Fecal samples—gut microbiome, oxidized guanine species	X	X	X	X
Blood samples—serum inflammatory cytokines, Plasma Vitamin K, oxidized guanine species	X	X	X	X
Acceptability—Food Acceptability Questionnaire		X		X
Demographics	X			
Dietary Habits and Colon Cancer Beliefs Survey	X			X

X indicates the outcome, measure, or specimen was collected at that time point.

**Table 2 nutrients-11-02349-t002:** Characteristics of participants and non-participants in a randomized controlled crossover high green leafy vegetable dietary intervention.

	Participants	Non-Participants	
	(*n* = 50)	(*n* = 128)	
	------- Mean ^1^ (SD) ------		*p*
RM servings per week	10.3 (5.1)	8.1 (7.2)	0.050
GLV servings per week	0.21 (0.25)	0.28 (0.45)	0.215
Age (years)	48 (13)	45 (12)	0.100
Body Mass Index (kg/m^2^)	36.2 (4.9)	31.2 (9.0)	<0.0001
	-------- N ^2^ (%) --------		*p*
Gender			0.211
Male	19 (38%)	36 (28.1%)	
Female	31 (62%)	92 (71.9%)	
Race			0.293
Asian	0 (0%)	4 (3.1%)	
African-American	10 (20%)	25 (19.5%)	
White	40 (80%)	94 (73.4%)	
More than one race	0 (0%)	5 (3.9%)	
Education			0.060
Less than bachelor’s degree	7 (14%)	39 (30.5%)	
Bachelor’s degree	19 (38%)	28 (21.9%)	
Master’s degree	13 (26%)	31 (24.2%)	
Doctoral/Professional degree	11 (22%)	30 (23.4%)	
Marital Status			0.998
Single	12 (24%)	31 (24.2%)	
Married	28 (56%)	71 (55.5%)	
Widowed/Divorced/Separated	10 (20%)	26 (20.3%)	

^1^ independent sample *t*-tests for continuous variables. ^2^ chi-square tests for categorical variables. RM, red meat; GLVs, green leafy vegetables; SD, Standard Deviation.

**Table 3 nutrients-11-02349-t003:** Adherence results as measured by 24-h dietary recall data generated from the Automated Self-Administered 24-Hour Dietary Assessment tool (ASA24) in a randomized controlled crossover high green leafy vegetable dietary intervention.

	T0	T4	Change (T4–T0)		T8	T12	Change (T12–T8)	
Immediate Group	------- Mean (SD) ------			*p* ^1^	------- Mean (SD) ------			*p* ^1^
Vitamin K (mcg)	195.5 (264.1)	703.6 (752.5)	508.1 (854.5)	**0.009**	221.2 (203.5)	252.9 (238.6)	31.7 (319.7)	0.639
Dark Green Vegetables (cup eq.)	0.27 (0.36)	0.86 (0.92)	0.58 (1.02)	**0.011**	0.35 (0.41)	0.48 (0.50)	0.13 (0.69)	0.391
Red Meat (28 g eq.)	1.80 (1.87)	2.50 (2.45)	0.70 (2.92)	**0.262**	1.38 (1.21)	1.72 (1.83)	0.34 (2.01)	0.418
Delayed Group								
Vitamin K (mcg)	189.1 (162.9)	294.3 (395.5)	105.2 (389.3)	0.208	247.6 (380.6)	537.3 (471.8)	289.7 (373.5)	**0.001**
Dark Green Vegetables (cup eq.)	0.30 (0.37)	0.38 (0.47)	0.08 (0.63)	0.554	0.37 (0.567)	0.85 (0.79)	0.48 (0.76)	**0.006**
Red Meat (28 g eq.)	1.81 (1.82)	1.55 (1.45)	−0.26 (2.07)	0.548	0.977 (1.057)	2.14 (1.96)	1.17 (2.34)	**0.026**

The bold was to delineate which *p* values reflected the intervention period for that group; ^1^ paired sample *t*-test.

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
