# Peer review of "Primary Outcomes of a Randomized Controlled Crossover Trial to Explore the Effects of a High Chlorophyll Dietary Intervention to Reduce Colon Cancer Risk in Adults: The Meat and Three Greens (M3G) Feasibility Trial"

_nutrients, 2019, doi:10.3390/nu11102349_

Round 1
Reviewer 1 Report
The article is interesting for the journal and deals with an interesting issue.
There are some concerns that need to be addressed by the authors to further improve the quality of the manuscript.
- Please, the authors should clarify if the intervention is addressed to colon cancer (CC) or colorectal cancer (CRC). If the study is about CRC risk please modify the title of your manuscript.
- Please provide a full description of the acronyms when cited the first time, even in the abstract (e.g. BMI, CRC).
- Did the authors consider cooking methods? Boiling, stir-frying, steaming can affect the content of many micronutrients and bioactive compounds. Which kinds of cooking methods were applied?
- Page 12 row #254-256 "While diverticulosis is a heterogeneous disease which is often exacerbated by dietary fiber, we chose not to exclude individuals with this condition, especially since they are at increased risk of developing CRC". I suggest the authors, to check this phrase considering to use "diverticulitis" instead of "diverticulosis". Please add some references.
- Discussion paragraph. I suggest the authors to consider new evidence in the field of colon, rectal, and colorectal cancer prevention. I suggest to consider these papers to better argument the discussion:
1) Gianfredi, V.; Nucci, D.; Salvatori, T.; Dallagiacoma, G.; Fatigoni, C.; Moretti, M.; Realdon, S. Rectal Cancer: 20% Risk Reduction Thanks to Dietary Fibre Intake. Systematic Review and Meta-Analysis. Nutrients 2019, 11, 1579.
2) World Cancer Research Fund International/American Institute for Cancer Research. Countinous Update Project Report: Diet, Nutrition, Physical Activity and Colorectal Cancer; World Cancer Research Fund International/American Institute for Cancer Research: London, UK, 2017.
3) Gianfredi, V.; Salvatori, T.; Villarini, M.; Moretti, M.; Nucci, D.; Realdon, S. Is dietary fibre truly protective against colon cancer? A systematic review and meta-analysis. Int. J. Food Sci. Nutr. 2018, 69, 904–915.
Author Response
Reviewer 1
The article is interesting for the journal and deals with an interesting issue.
There are some concerns that need to be addressed by the authors to further improve the quality of the manuscript.
Thank you for your interest in this study and comments to improve the quality of this manuscript. All responses are reported below in bold.
- Please, the authors should clarify if the intervention is addressed to colon cancer (CC) or colorectal cancer (CRC). If the study is about CRC risk please modify the title of your manuscript.
We took great care to only mention colon cancer in the introduction and methods as this is the focus of the paper; however, when addressing other lifestyle interventions in the discussion, we included CRC studies and acknowledged them as such since they are relevant and inclusive of CC.
- Please provide a full description of the acronyms when cited the first time, even in the abstract (e.g. BMI, CRC).
We apologize for this oversight and have described acronyms appropriately.
- Did the authors consider cooking methods? Boiling, stir-frying, steaming can affect the content of many micronutrients and bioactive compounds. Which kinds of cooking methods were applied?
Because chlorophyll was our target nutrient and is retained at high heat, we did not collect data on specific cooking methods. This limitation is acknowledged in our Discussion.
- Page 12 row #254-256 "While diverticulosis is a heterogeneous disease which is often exacerbated by dietary fiber, we chose not to exclude individuals with this condition, especially since they are at increased risk of developing CRC". I suggest the authors, to check this phrase considering to use "diverticulitis" instead of "diverticulosis". Please add some references.
We have changed the wording and added a citation as follows: Diverticular disease is a heterogeneous disease which is often exacerbated by dietary fiber, we chose not to exclude individuals with this condition, especially since they are at increased risk of developing CC or colorectal cancer (CRC).
- Discussion paragraph. I suggest the authors to consider new evidence in the field of colon, rectal, and colorectal cancer prevention. I suggest to consider these papers to better argument the discussion:
1) Gianfredi, V.; Nucci, D.; Salvatori, T.; Dallagiacoma, G.; Fatigoni, C.; Moretti, M.; Realdon, S. Rectal Cancer: 20% Risk Reduction Thanks to Dietary Fibre Intake. Systematic Review and Meta-Analysis. Nutrients 2019, 11, 1579.
2) World Cancer Research Fund International/American Institute for Cancer Research. Countinous Update Project Report: Diet, Nutrition, Physical Activity and Colorectal Cancer; World Cancer Research Fund International/American Institute for Cancer Research: London, UK, 2017.
3) Gianfredi, V.; Salvatori, T.; Villarini, M.; Moretti, M.; Nucci, D.; Realdon, S. Is dietary fibre truly protective against colon cancer? A systematic review and meta-analysis. Int. J. Food Sci. Nutr. 2018, 69, 904–915.
The following has been added to the discussion: Moreover, a recent meta-analysis of 21 cohort and case-control studies indicated significantly lower risk of CC in men and women with the highest levels of dietary fiber consumption [35].
Reviewer 2 Report
“Primary outcomes of randomized controlled crossover trial to explore the effects of a high chlorophyll dietary intervention to reduce colon cancer risk in adults: The Meat and Three Greens (M3G) Feasibility Trial”
Authors: Fruge AD, Smith KS, Riviere AJ, Denmark-Wahnefried W, Arthur AE, Murrah WM, Morrow CD, Arnold RD, Braxton-Lloyd K
Summary:
High meat diet, obesity and low physical activity all increase risk for colon cancer. Especially intake of heme may act as a principal carcinogen in meat. Previous preclinical studies on rodents have shown that intake of chlorophyll significantly prevents cytotoxic damage in the colon crypts and may reduce risk of colon cancer. As changing colon cancer risk-patients dietary patterns towards a meat reduced diet is difficult, the authors investigated in a randomized controlled crossover study over 12 weeks the acceptance of a high green leafy vegetable intake diet instead.
Comments:
Major Points:
As indeed reduction of meat consumption remains difficult with majority of colon cancer risk and obese patients, it is a very interesting approach to enhance uptake of chlorophyll intake to exert protective effects in the colon. However, just to present the feasibility of the study as a manuscript is too little for a scientific article! It would give the article more scientific value if also clinically relevant parameters would be included. Indeed, the authors state they investigated inflammatory markers, DNA damage and change of gut microbiome composition but these are not part of the presented paper as they should be!
Minor points:
Please change US customary system of measurement (such as “ounce”) to international system of measurement (metrics). It would be further interesting to hypothesize on the interaction between, microbes, heme and chlorophyll as heme and chlorophyll are very similar in structure and differ only by the Fe and Mg ion.
Author Response
Reviewer 2
Summary:
High meat diet, obesity and low physical activity all increase risk for colon cancer. Especially intake of heme may act as a principal carcinogen in meat. Previous preclinical studies on rodents have shown that intake of chlorophyll significantly prevents cytotoxic damage in the colon crypts and may reduce risk of colon cancer. As changing colon cancer risk-patients dietary patterns towards a meat reduced diet is difficult, the authors investigated in a randomized controlled crossover study over 12 weeks the acceptance of a high green leafy vegetable intake diet instead.
Comments:
Major Points:
As indeed reduction of meat consumption remains difficult with majority of colon cancer risk and obese patients, it is a very interesting approach to enhance uptake of chlorophyll intake to exert protective effects in the colon. However, just to present the feasibility of the study as a manuscript is too little for a scientific article! It would give the article more scientific value if also clinically relevant parameters would be included. Indeed, the authors state they investigated inflammatory markers, DNA damage and change of gut microbiome composition but these are not part of the presented paper as they should be!
Thank you for the commendation of our approach. We, too, are excited to see the results of all biological specimens and will report them in their entirety once they have all analyses are complete. The primary outcomes of this study as reported a priori on clinicaltrials.gov were feasibility; this is a required step in behavioral interventions as the investigators must prove the study can be conducted and the participants are actually recruited, stay in the study, and adhere to the protocol.
Minor points:
Please change US customary system of measurement (such as “ounce”) to international system of measurement (metrics).
Ounce equivalents have been changed to 28 gram equivalents.
It would be further interesting to hypothesize on the interaction between, microbes, heme and chlorophyll as heme and chlorophyll are very similar in structure and differ only by the Fe and Mg ion.
This has been addressed in preliminary studies by the group in the Netherlands and extra verbiage has been added to the introduction: Interestingly, spinach and chlorophyll prevent cytotoxicity and damage to colonocytes in vivo by the binding of heme between chlorophyll molecules.
Reviewer 3 Report
Authors of this research article presented results obtained from feasibility study demonstrating the green leafy vegetable diet intervention in adults at high risk colon cancer. This is a well performed study and a well written article. It would be interesting to see analysis of biological data collected.
Author Response
Reviewer 3
Authors of this research article presented results obtained from feasibility study demonstrating the green leafy vegetable diet intervention in adults at high risk colon cancer. This is a well performed study and a well written article. It would be interesting to see analysis of biological data collected.
Thank you for your comments and interest in our study. As stated in the first response to reviewer #2, the primary aim of this pilot study was feasibility and this is the focus of this specific paper; therefore, we cover issues related to feasibility and adherence in necessary detail. We, too, look forward to the biological data and are eager for the analyses to be complete.
Round 2
Reviewer 1 Report
The authors have done excellent work in addressing the feedback provided. I have no further comments to offer.
Reviewer 2 Report
Thank you, the authors have answered our questions.